# Amplifiers of selection for the Moran process with both Birth-death and death-Birth updating

**Jakub Svoboda**[1], **Soham Joshi**[1], **Josef Tkadlec**[2], **Krishnendu Chatterjee**[1]*

**1** IST Austria, Klosterneuburg, Austria, **2** Computer Science Institute, Charles University, Prague, Czech Republic

* krishnendu.chatterjee@ist.ac.at

**Data Availability Statement:** Code for the figures and the computational experiments is available from the Figshare repository: https://figshare.com/s/4e08d78c892749f84201.

## Abstract

Populations evolve by accumulating advantageous mutations. Every population has some spatial structure that can be modeled by an underlying network. The network then influences the probability that new advantageous mutations fixate. Amplifiers of selection are networks that increase the fixation probability of advantageous mutants, as compared to the unstructured fully-connected network. Whether or not a network is an amplifier depends on the choice of the random process that governs the evolutionary dynamics. Two popular choices are Moran process with Birth-death updating and Moran process with death-Birth updating. Interestingly, while some networks are amplifiers under Birth-death updating and other networks are amplifiers under death-Birth updating, so far no spatial structures have been found that function as an amplifier under both types of updating simultaneously. In this work, we identify networks that act as amplifiers of selection under both versions of the Moran process. The amplifiers are robust, modular, and increase fixation probability for any mutant fitness advantage in a range $r \in (1, 1.2)$. To complement this positive result, we also prove that for certain quantities closely related to fixation probability, it is impossible to improve them simultaneously for both versions of the Moran process. Together, our results highlight how the two versions of the Moran process differ and what they have in common.

## Author summary

The long-term fate of an evolving population depends on its spatial structure. Amplifiers of selection are spatial structures that enhance the probability that a new advantageous mutation propagates through the whole population, as opposed to going extinct. Many amplifiers of selection are known when the population evolves according to the Moran Birth-death updating, and several amplifiers are known for the Moran death-Birth updating. Interestingly, none of the spatial structures that work for one updating seem to work for the other one. Nevertheless, in this work we identify spatial structures that function as amplifiers of selection for both types of updating. We also prove two negative

**Funding:** J.S., S.J., and K.C were supported by European Research Council (ERC) CoG 863818 (ForM-SMArt). J.T was supported by Center for Foundations of Modern Computer Science (Charles University project UNCE/SCI/004) and by the project PRIMUS/24/SCI/012 from Charles University. The funders had no role in study design, data collection and analysis, decision to publish, or preparation of the manuscript.

**Competing interests:** The authors have declared that no competing interests exist.

results that suggest that stumbling upon such spatial structures by pure chance is unlikely.

## Introduction

Moran process is a classic stochastic process that models natural selection in populations of asexually reproducing individuals, especially when new mutations are rare [1, 2]. It is commonly used to understand the fate of a single new mutant, as it attempts to invade a population of indistinguishable residents. Eventually, the new mutation will either fixate on the whole population, or it will go extinct. It is known that when the invading mutant has relative fitness advantage $r > 1$ as compared to the residents, this fixation probability tends to a positive constant $1 - 1/r$ as the population size $N$ grows large.

On spatially structured populations, fixation probability of an invading mutant can both increase or decrease. In the framework of evolutionary graph theory [3, 4], the spatial structure is represented by a graph (network) in which nodes (vertices) correspond to individual sites, and edges (connections) correspond to possible migration patterns. Each edge is assigned a weight that represents the strength of the connection. Such network-based spatial structures can represent island models, metapopulations, lattices, as well as other arbitrarily complex structures [5–9]. Spatial structures that increase the fixation probability of a randomly occurring advantageous mutant beyond the constant $1 - 1/r$ are called amplifiers of selection [10]. The logic behind the name is that living on such a structure effectively amplifies the fitness advantage that the mutants has, as compared to living on the unstructured (well-mixed) population. Identifying amplifiers is desirable, since they could potentially serve as tools in accelerating the evolutionary search, especially when new mutations are rare [11, 12].

When run on a spatial structure, Moran process can be implemented in two distinct versions. They are called Moran Birth-death process and Moran death-Birth process. In the Moran Birth-death process, first an individual is selected for reproduction with probability proportional to its fitness, and the offspring then replaces a random neighbor. In contrast, in the Moran death-Birth process, first a random individual dies and then its neighbors compete to fill up the vacant site (see Fig 1). Both the Moran Bd-updating [1, 3, 13] and the Moran dB-updating [14–17] have been studied extensively. While essentially identical on the unstructured population, the two versions of the process yield different results when run on most spatial structures [18–20].

In the world of the Bd-updating, amplifiers are ubiquitous [12, 21–24]. Almost all small spatial structures function as amplifiers of selection [21]. A prime example of an amplifier under the Bd-updating is the Star graph, which improves the mutant fixation probability to roughly $1 - 1/r^2$ [19, 25–27]. In particular, when $r = 1 + \varepsilon$, this is approximately a two-fold increase over the baseline value $1 - 1/r$ given by the unstructured population. Moreover, certain large spatial structures function as so-called superamplifiers, that is, they increase the mutant fixation probability arbitrarily close to 1, even when the mutant has only negligible fitness advantage $r = 1 + \varepsilon$ [28]. Many other superamplifiers are known, including Incubators [29], or Selection Reactors [30].

In contrast, in the world of dB-updating, only a handful of amplifiers are known [31]. Perhaps the most prominent examples are the Fan graphs (see Fig 2) that increase the fixation probability of near-neutral mutants by a factor of up to 1.5 [32]. Interestingly, all dB-amplifiers are necessarily transient, meaning that the provided amplification effect disappears when the mutant fitness advantage exceeds a certain threshold [33]. In particular, large Fan graphs

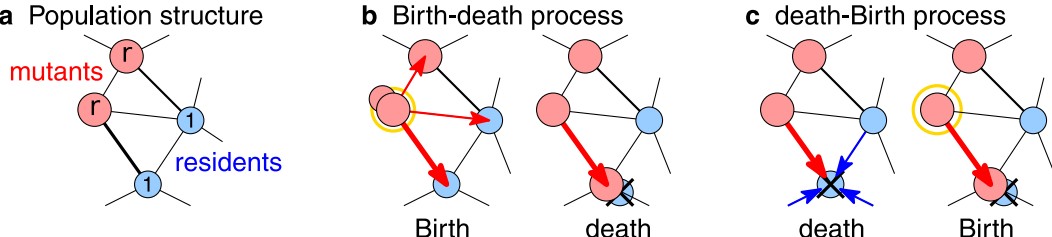

**Fig 1. Moran Birth-death and death-Birth processes on a population structure. a,** Each node is occupied by a resident with fitness 1 (blue), or a mutant with fitness $r \geq 1$ (red). Thicker edges denote higher edge weights (stronger interactions). **b,** In Moran Birth-death process, a random individual reproduces, and the produced offspring migrates along a random edge. **c,** In Moran death-Birth process, a random individual dies, and the vacancy is filled by a random neighbor. In both cases, edges with higher weight are selected more often, and fitness plays a role in the Birth step but not in the death step.

increase the fixation probability of the invading mutants for $r \in (1, \varphi)$, where $\varphi \approx 1.618$ is the golden ratio, but decrease it when $r > \varphi$ [32].

Unfortunately, the Fan graphs do not function as amplifiers when we instead consider them under Bd-updating (see Fig 2). This is unexpected, since amplification in the Bd-world is so pervasive. And it begs a question. Do there exist spatial structures that function as amplifiers both under the Bd-updating and under the dB-updating? That is, do there exist structures for which the amplification effect is robust with respect to the seemingly arbitrary choice of which version of the Moran process we decide to run?

In this work, we first show three negative results that indicate that the requirements for Bd-amplification and dB-amplification are often conflicting. First, we show that known amplifiers of selection under the Bd-updating are suppressors of selection for the dB-updating and vice versa. Second, we prove that simultaneous Bd- and dB-amplification is impossible under neutral drift ($r = 1$) when the initial mutant location is fixed to a specific starting node. Third, we define a quantity that corresponds to the probability of "mutants going extinct immediately". We then prove that, roughly speaking, no graph improves this quantity as compared to the complete graph under both Bd- and dB-updating. Thus, improving fixation probability under both Bd- and dB-udpating as compared to the complete graph might seem unlikely.

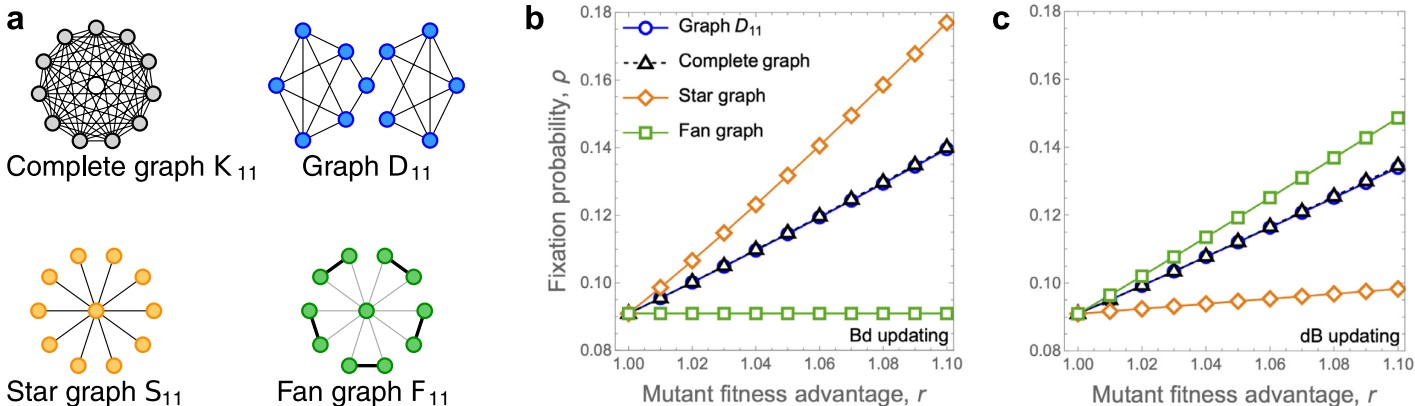

**Fig 2. Known amplifiers are suppressors for the other process. a,** We consider four graphs on $N = 11$ nodes, namely the Complete graph $K_{11}$, the star graph $S_{11}$, the Fan graph $F_{11}$, and the smallest known undirected amplifier $D_{11}$ (see [17]). **b,** Under Bd-updating, the only amplifier for $r \in \{1.01, \ldots, 1.1\}$ is the Star graph $S_{11}$. **c,** Under dB-updating, the only amplifier for $r \in \{1.01, \ldots, 1.1\}$ is the Fan graph $F_{11}$. Values computed by numerically solving the underlying Markov chains.

Despite those negative results, we identify a class of population structures that function as amplifiers of selection under both Birth-death and death-Birth updating, for any mutation that grants a relative fitness advantage $r \in (1, 1.2)$. We also present numerical computation that illustrates that the amplification strength is substantial.

## Model

Here we formally introduce the terms and notation that we use later, such as the evolutionary dynamics of Moran Birth-death and Moran death-Birth process, the fixation probability, and the notion of an amplifier.

### Population structure

The spatial structure of the population is represented as a graph (network), denoted $G_N = (V, E)$, where $V$ is a set of $N$ nodes (vertices) of $G_N$ that represent individual sites, and $E$ is a set of edges (connections) that represent possible migration patterns for the offspring. The edges are undirected (two-way) and may be weighted to distinguish stronger interactions from the weaker ones, see Fig 1a. The weight of an edge between nodes $u$ and $v$ is denoted $w(u, v)$. If all edge weights are equal to 1 we say that the graph is *unweighted*. At any given time, each site is occupied by a single individual, who is either a resident with fitness 1, or a mutant with fitness $r \geq 1$. The fitness of an individual at node $u$ is denoted $f(u)$.

### Moran process

Moran process is a classic discrete-time stochastic process that models the evolutionary dynamics of selection in a population of asexually reproducing individuals. Initially, each node is occupied either by a resident or by a mutant. As long as both mutants and residents co-exist in the population, we perform discrete time steps that change the state of (at most) one node at a time.

There are two versions of the Moran process (see Fig 1). In the *Moran Birth-death process*, we first select an individual to reproduce (randomly, proportionally to the fitness of the individual), and then the offspring migrates along one adjacent edge (randomly, proportionally to the weight of that edge) to replace the neighbor. Formally, denoting by $F = \sum_u f(u)$ the total fitness of the population, node $u$ gets selected for reproduction with probability $f(u)/F$, and then it replaces a neighbor $v$ with probability $p_{u \to v} = w(u, v)/\sum_{v'} w(u, v')$.

In contrast, in the *Moran death-Birth process*, we first select an individual to die (uniformly at random), and then the neighbors compete to fill in the vacancy (randomly, proportionally to the edge weight and the fitness of the neighbor). Formally, node $v$ dies with probability $1/N$ and it gets replaced by a node $u$ with probability $p_{u \to v} = f(u) \cdot w(u, v)/(\sum_{u'} f(u') \cdot w(u', v))$. We note that in both versions we capitalize the word "Birth" to signify that fitness plays a role in the birth step (and not in the death step).

### Fixation probability and amplifiers

If the graph $G_N$ that represents the population structure is connected then the Moran process eventually reaches a "homogeneous state", where either all nodes are occupied by mutants (we say that mutants *fixated*), or all nodes are occupied by residents (we say that mutants *went extinct*). Given a graph $G_N$, a mutant fitness advantage $r \geq 1$, and a set $S \subseteq V$ of nodes initially occupied by mutants, we denote by $\rho_r^{\text{Bd}}(G_N, S)$ the *fixation probability*, that is, the probability that mutants eventually reach fixation, under Moran Birth-death process. We are particularly interested in the fixation probability of a single mutant who appears at a node selected

uniformly at random. We denote this *fixation probability under uniform initialization* by $\rho_r^{\text{Bd}}(G_N) = \frac{1}{N}\sum_{v \in V} \rho_r^{\text{Bd}}(G_N, \{v\})$. We define $\rho_r^{\text{dB}}(G_N, S)$ and $\rho_r^{\text{dB}}(G_N)$ analogously.

In this work we focus on population structures that increase the fixation probability of invading mutants. The base case is given by an unweighted complete graph $K_N$ that includes all edges and represents an unstructured, well-mixed population. It is known [4, 20, 21] that

$$\rho_r^{\text{Bd}}(K_N) = \frac{1 - \dfrac{1}{r}}{1 - \dfrac{1}{r^N}} \quad \text{and} \quad \rho_r^{\text{dB}}(K_N) = \frac{N-1}{N} \cdot \frac{1 - \dfrac{1}{r}}{1 - \dfrac{1}{r^{N-1}}}. \tag{1}$$

Given a graph $G_N$ and a mutant fitness advantage $r \geq 1$, we say that $G_N$ is a *Bd$_r$-amplifier* if $\rho_r^{\text{Bd}}(G_N) > \rho_r^{\text{Bd}}(K_N)$. We define dB$_r$ amplifiers analogously, that is, as those graphs $G_N$ that satisfy $\rho_r^{\text{dB}}(G_N) > \rho_r^{\text{dB}}(K_N)$. Similarly, *suppressors* are graphs that decrease the fixation probability as compared to the complete graph.

## Results

First, we present three negative results that illustrate that the two worlds of Birth-death and death-Birth updating often present contradictory requirements when it comes to enhancing the fixation probability of a single newly occurring mutant. Nevertheless, as our main contribution in the positive direction, we then present population structures that are both Bd$_r$-amplifiers and dB$_r$-amplifiers for a range of mutant fitness advantages $r \in (1, 1.2)$.

### Negative results

In this section, we present results that suggest that finding simultaneous Bd$_r$- and dB$_r$- amplifiers is not easy. First, we show empirically that known amplifiers for one process are suppressors for the other process. Second, we show that in the neutral regime ($r = 1$), any fixed vertex is a "good" starting vertex for the mutant in at most one of the two processes. Finally, we show that for any starting vertex, the chance of not dying immediately can be enhanced in at most one of the two processes (see below for details).

**Known amplifiers for one process.** In this section we examine spatial structures that are known to amplify under one of the two versions of the Moran process, in order to see whether they amplify under the other version of the Moran process.

First, we consider the smallest known unweighted dB-amplifier [17], which is a certain graph on $N = 11$ nodes (see Fig 2). We call the graph $D_{11}$. The graph $D_{11}$ is an extremely weak dB$_r$-amplifier in a range of approximately $r \in (1, 1.00075)$, where it increases the fixation probability by a factor less than $1.0000001\times$ (see [17], Fig 1). For $r \in (1.01, 1.1)$ the graph $D_{11}$ appears to function as a very slight suppressor under both dB-updating and Bd-updating. In particular, at $r = 1.1$ we obtain $\rho_r^{\text{Bd}}(D_{11})/\rho_r^{\text{Bd}}(K_{11}) \doteq 0.996$ and $\rho_r^{\text{dB}}(D_{11})/\rho_r^{\text{dB}}(K_{11}) \doteq 0.997$.

Next, we examine the star graph $S_{11}$ on 11 vertices which, to our knowledge, is the strongest unweighted amplifier for Bd-updating at this population size. The Star graph is a clear Bd$_r$-amplifier for $r \in (1.01, 1.1)$, but an equally clear dB$_r$-suppressor in that range.

The situation is reversed for the Fan graph $F_{11}$ [32]. While the Fan graph clearly functions as an amplifier under the dB-updating when $r \in (1.01, 1.1)$, it lags behind the baseline given by the complete graph under the Bd-updating.

**Neutral regime ($r = 1$).** The second negative result pertains to the case of neutral mutations ($r = 1$). Recall that $\rho_r^{\text{Bd}}(G_N, v)$ and $\rho_r^{\text{dB}}(G_N, v)$ denote the fixation probabilities when the initial mutant appears at node $v$. The following theorem states that for neutral mutations

($r = 1$), no initial mutant node increases the fixation probability both for Birth-death and death-Birth updating.

**Theorem 1**. *Let $G_N$ be a graph and $v$ an initial mutant node. Then at least one of the following is true*:

1. $\rho_{r=1}^{\text{Bd}}(G_N, v) < \rho_{r=1}^{\text{Bd}}(K_N)$; *or*

2. $\rho_{r=1}^{\text{dB}}(G_N, v) < \rho_{r=1}^{\text{dB}}(K_N)$; *or*

3. $\rho_{r=1}^{\text{Bd}}(G_N, v) = \rho_{r=1}^{\text{Bd}}(K_N)$ *and* $\rho_{r=1}^{\text{dB}}(G_N, v) = \rho_{r=1}^{\text{dB}}(K_N)$.

The idea behind the proof is that for neutral evolution there are explicit formulas for fixation probabilities $\rho_r^{\text{Bd}}(G_N, v)$ and $\rho_r^{\text{dB}}(G_N, v)$ on any undirected graph $G_N$ [34, 35]. The result then follows by applying Cauchy-Schwarz inequality. See Supplementary Information for details. In Supplementary Information, we also note that Theorem 1 does not generalize to the case when instead of having one initial mutant node we start with an initial subset $S$ of $k \geq 2$ nodes occupied by mutants.

**Immediate extinction and forward bias.**   In order to present our third and final negative result, we need to introduce additional notions and notation. When tracking the evolutionary dynamics on a given graph $G_N$ with a given mutant fitness advantage $r \geq 1$, it is often useful to disregard the exact configuration of which nodes are currently occupied by mutants, and only look at *how many* nodes are occupied by mutants.

One example of this is the celebrated Isothermal theorem [3] which states that once $N$ and $r$ are fixed, the fixation probability under the Moran Birth-death process on any regular graph is the same. Here, a graph is *regular* if each node has the same total weight of adjacent edges. Examples of regular graphs include the complete graph, the cycle graph, or any grid graph with periodic boundary condition.

The intuition behind the proof of the Isothermal theorem is that for any regular graph $R_N$, the Moran Birth-death process can be mapped to a random walk that tracks just the number of mutants, instead of their exact positions on the graph. It can be shown that this random walk has a constant forward bias, that is, the probabilities $p^+$ (resp. $p^-$) that the size of the mutant subpopulation increases (resp. decreases) satisfy $p^+/p^- = r$, for any number of mutants in any particular mutant-resident configuration. A natural approach to construct amplifiers is thus to construct graphs for which this forward bias satisfies an inequality $p^+/p^- \geq r$ for the Moran Birth-death process and an analogous inequality for the Moran death-Birth process. Our final negative result shows that this goal can not be achieved already in the first step.

Formally, consider the Moran Birth-death process on a graph $G_N$ with a single initial mutant placed at node $u$. Let $\gamma_r^{\text{Bd}}(G_N, u)$ be the probability that the first reproduction event that changes the size of the mutant subpopulation is the initial mutant reproducing (as opposed to the initial mutant being replaced by one of its neighbors). In other words, $\gamma_r^{\text{Bd}}(G_N, u)$ is the probability that the first step that changes the configuration of the mutants does *not* eliminate the initial mutant, leaving the options of later mutant extinction or mutant fixation.

For the complete graph $K_N$ (and any single mutant node) it is not hard to show that $\gamma_r^{\text{Bd}}(K_N) = \gamma_r^{\text{Bd}}(K_N, u) = r/(r + 1)$ for any node $u$. Moreover, by a slight extension of the Isothermal theorem, we have $\gamma_r^{\text{Bd}}(R_N, u) = r/(r + 1)$ for any regular graph $R_N$ and any node $u$. For Moran death-Birth process, we define $\gamma_r^{\text{dB}}(G_N, u)$ and $\gamma_r^{\text{dB}}(K_N)$ analogously. To construct a graph that is both a Bd- and a dB-amplifier, a natural approach is to look for a graph and an initial mutant node $u$ such that $\gamma_r^{\text{Bd}}(G_N, u) > \gamma_r^{\text{Bd}}(K_N)$ and $\gamma_r^{\text{dB}}(G_N, u) > \gamma_r^{\text{dB}}(K_N)$. However, the following theorem states that no such graphs exist.

**Theorem 2**. *Let $G_N$ be a graph, $u$ an initial mutant node, and $r \geq 1$. Then at least one of the following is true*:

1. $\gamma_r^{\mathrm{Bd}}(G_N, u) < \gamma_r^{\mathrm{Bd}}(K_N)$; *or*

2. $\gamma_r^{\mathrm{dB}}(G_N, u) < \gamma_r^{\mathrm{dB}}(K_N)$; *or*

3. $\gamma_r^{\mathrm{Bd}}(G_N, u) = \gamma_r^{\mathrm{Bd}}(K_N)$ *and* $\gamma_r^{\mathrm{dB}}(G_N, u) = \gamma_r^{\mathrm{dB}}(K_N)$.

The proof relies on the notion of the temperature of a node. Formally, given a graph $G_N = (V, E)$ we first define a (weighted) *degree* of a node $v$ as $\deg(v) = \sum_{v': (v, v') \in E} w(v, v')$. Then, given a node $u$, we define its *temperature* $T(u)$ as

$$T(u) = \sum_{v:(v,u) \in E} \frac{w(v, u)}{\deg(v)}.$$

The temperature of a node represents the rate at which the node is being replaced by its neighbors in the Moran Birth-death process when $r = 1$. Nodes with high temperature are replaced often, whereas nodes with low temperature are replaced less frequently. Building on this, it is straightforward to show that if a node $u$ has above-average temperature, then $\gamma_r^{\mathrm{Bd}}(G_N, u) < \gamma_r^{\mathrm{Bd}}(K_N)$, that is, in Moran Birth-death process with a single mutant at $u$ the forward bias is lower than the forward bias on a complete graph. To complete the proof, we then show that for any node $u$ with below-average temperature, we have $\gamma_r^{\mathrm{dB}}(G_N, u) < \gamma_r^{\mathrm{dB}}(K_N)$. Our proof of the latter claim uses Jensen's inequality for a certain concave function. See Supplementary Information for details.

## Positive result

Despite the above negative results, in this section we identify population structures $A_N$ that substantially amplify the fixation probability under both Birth-death updating and death-Birth updating when the number $N$ of nodes is sufficiently large.

The structures $A_N$ are composed of two large chunks $A^{\mathrm{Bd}}$ and $A^{\mathrm{dB}}$ that are connected by a single edge, see Fig 3a for an illustration. The chunk $A^{\mathrm{dB}}$ is a Fan graph [32], which is to our knowledge the strongest currently known dB-amplifier. The chunk $A^{\mathrm{Bd}}$ could be any of the many strong Bd-amplifiers. For definiteness, in Fig 3a we use a Fan-like structure with $a$ nodes in a central hub and $b$ blades of two nodes each surrounding it. The single connecting edge has a very low edge weight so that the two chunks interact only rarely. For population size $N = 1001$, the resulting weighted graph is both a Bd$_r$-amplifier and a dB$_r$-amplifier for any $r \in (1, 1.09)$, see Fig 3b.

Similarly, we identify large population structures that serve as both Bd$_r$-amplifiers and dB$_r$-amplifiers for any $r \in (1, 1.2)$.

**Theorem 3 (Simultaneous Bd- and dB-amplifier)**. *For every large enough population size $N$ there exists a graph $A_N$ such that for all $r \in (1, 1.2)$ we have*

$$\rho_r^{\mathrm{Bd}}(A_N) > \rho_r^{\mathrm{Bd}}(K_N) \quad and \quad \rho_r^{\mathrm{dB}}(A_N) > \rho_r^{\mathrm{dB}}(K_N).$$

In what follows we provide intuition about the proof of Theorem 3. The fully rigorous proof is relegated to Supplementary Information. Let $e$ be the edge connecting the two chunks, $u$ its endpoint in $A^{\mathrm{Bd}}$, and $v$ its endpoint in $A^{\mathrm{dB}}$.

First, observe that since $e$ has a low weight, the two chunks evolve mostly independently. This means that, with high probability, each chunk resolves to a homogeneous state in between any two interactions across the chunks. In particular, if the initial mutant appears in the chunk

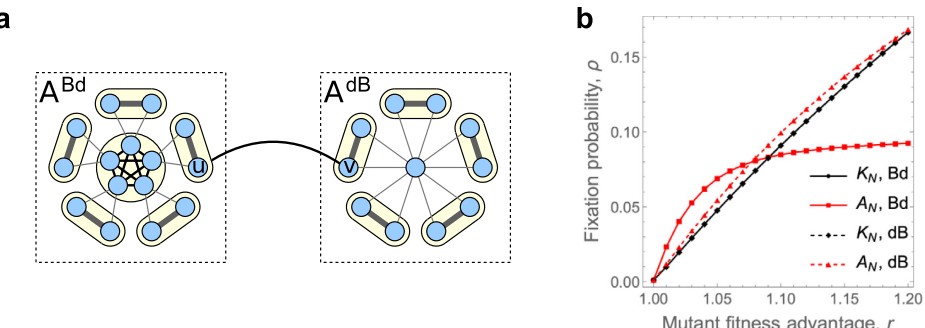

**Fig 3. Simultaneous Bd- and dB-amplifier $A_N$. a,** The graph $A_N$ is composed of two large chunks $A^{\text{Bd}}$ and $A^{\text{dB}}$ that are connected by a single edge. The chunk $A^{\text{dB}}$ is a Fan graph with $f$ nodes. The chunk $A^{\text{Bd}}$ is a fan-like graph with $a$ vertices in a central hub and $b$ blades of two nodes each. The total population size is $N = a + 2b + f$ (here $a = b = 5$, $f = 11$, and $N = 26$). The edge weights are defined such that different circled units within the chunks interact only rarely, and the chunks themselves interact even more rarely. **b,** Here we consider graph $A_N$ with population size $N = 1001$ and $(a, b, f) = (30, 85, 801)$. The fixation probabilities under Bd- and dB-updating are computed by numerically solving the underlying Markov chain. We find that the inequality $\rho_r^{\text{Bd}}(A_N) > \rho_r^{\text{Bd}}(K_N)$ is satisfied for $r \in (1, 1.09)$ and the inequality $\rho_r^{\text{dB}}(A_N) > \rho_r^{\text{dB}}(K_N)$ is satisfied for $r \in (1, 1.2)$. In particular, at $r = 1.05$ the ratios satisfy $\rho_r^{\text{Bd}}(A_N)/\rho_r^{\text{Bd}}(K_N) > 1.44$ and $\rho_r^{\text{dB}}(A_N)/\rho_r^{\text{dB}}(K_N) > 1.14$.

where it is favored (e.g. if it appears in the chunk $A^{\text{Bd}}$ when Bd-updating is run), the mutants fixate on that chunk with reasonable probability. If that occurs, we say that mutants are "half done".

Once the mutants are half done, the next relevant step occurs when the two chunks interact. There are two cases. Either a mutant at $u$ reproduces and the offspring migrates along $e$ to $v$, or a resident at $v$ reproduces and the offspring migrates along $e$ to replace the mutant at $u$. In both cases, the individual (mutant or resident) who "invades" the other half eventually either succeeds in spreading through that half, or they fail at doing that. If the latter occurs, we are back at the situation in which mutants are half done and the situation repeats. By bounding all the relevant probabilities, we show that once half done, mutants are overwhelmingly likely to fixate, as opposed to going extinct.

We highlight an interesting phenomenon that occurs in our proof. As we run the evolutionary dynamics, we can look at the flow along the connecting edge $e$. Thanks to the edge weights, it turns out that the direction of the flow along $e$ flips depending on whether we run the Moran Birth-death process or the Moran death-Birth process. In particular, under the Bd-updating the edge $e$ is used mostly in the direction from $u$ to $v$. That is, many individuals migrate from $u$ to $v$, whereas few individuals migrate from $v$ to $u$. Under dB-updating the situation reverses. That is, many individuals migrate from $v$ to $u$, whereas few of them migrate from $u$ to $v$. Thus, under the Bd-updating the $A^{\text{Bd}}$ chunk is effectively upstream of the chunk $A^{\text{dB}}$, whereas under the dB-updating the $A^{\text{dB}}$ chunk is effectively upstream of the chunk $A^{\text{Bd}}$. This asymmetry is a key factor that contributes to the fact that once the mutants are half done, they are likely to fixate on the whole graph (see Fig 4).

What remains in the proof is to balance out the sizes of the two chunks. For small $r > 1$, the strongest known dB-amplifiers are roughly $\frac{3}{2} \times$ stronger than the Complete graph (in terms of the fixation probability). Thus, in order to achieve amplification under dB-updating, we need the chunk $A^{\text{dB}}$ to take up at least 2/3 of the total population size. The chunk $A^{\text{Bd}}$ then takes up at most 1/3 of the total population size. In order to achieve Bd-amplification, fixation probability on $A^{\text{Bd}}$ under Bd-updating must therefore be at least 3× larger than that on the Complete graph. Interestingly, a Star graph is not strong enough to do that (for $r \approx 1$ and large

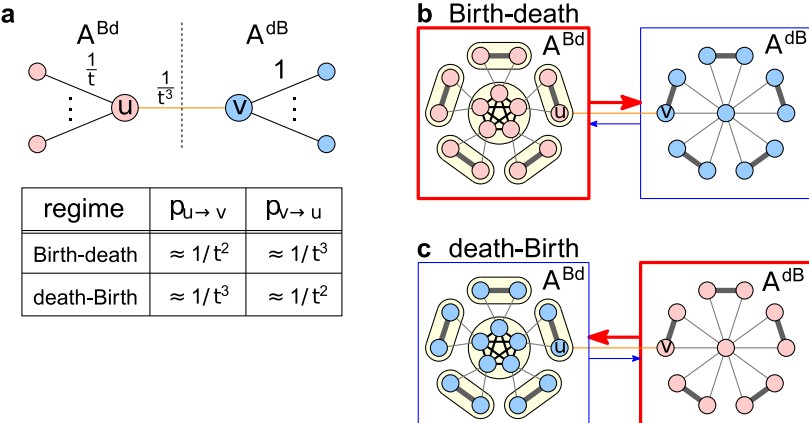

**Fig 4. Interactions between $A^{\text{Bd}}$ and $A^{\text{dB}}$. a,** The edge weights in the chunks $A^{\text{Bd}}$ (red) and $A^{\text{dB}}$ (blue) are shown as a function of $t$ (here $t \gg 1$ is large). The connecting edge has weight $1/t^3$, all other edges with endpoint $u$ have total weight $1/t$ and all other edges with endpoint $v$ have total weight 1. For each of two versions of the Moran process, the rates at which the offspring migrate from $u$ to $v$ and from $v$ to $u$ can be calculated and are listed in the table. **b,** Under Birth-death updating, the migration rate $p_{u \to v}$ from $u$ to $v$ is roughly $t\times$ larger than the migration rate $p_{v \to u}$ from $v$ to $u$, so the chunk $A^{\text{Bd}}$ is upstream of the chunk $A^{\text{dB}}$, and a mutant who has fixated over $A^{\text{dB}}$ is likely to fixate over $A^{\text{Bd}}$ too. **c,** In contrast, under death-Birth updating we have $p_{v \to u} \approx t \cdot p_{u \to v}$, hence the chunk $A^{\text{dB}}$ is upstream of $A^{\text{Bd}}$.

population size $N$ it is only roughly 2× stronger than the Complete graph), but sufficiently strong Bd-amplifiers do exist (e.g. any superamplifier).

## Discussion

Population structure has a profound impact on the outcomes of evolutionary processes and, in particular, on the probability that a novel mutation achieves fixation [3, 36]. Population structures that increase the fixation probability of beneficial mutants, when compared to the case of a well-mixed population, are known as amplifiers of selection.

Somewhat surprisingly, to tell whether a specific spatial structure is an amplifier or not, one needs to specify seemingly minor details of the evolutionary dynamics. The well-studied Moran process comes in two versions, namely Moran process with Birth-death updating and Moran process with death-Birth updating. While many spatial structures are amplifiers under the Bd-updating [21], only a handful of amplifiers under the dB-updating are known [31]. Moreover, none of the dB-amplifiers that we checked amplify under the Bd-updating.

In this work we help explain this phenomenon by proving mathematical results which illustrate that the two objectives of amplifying under the Bd-updating and amplifying under the dB-updating are often contradictory. Thus, one might be tempted to conclude that perhaps there are no population structures that amplify in both worlds, that is, regardless of the choice of the underlying dynamics (Bd or dB). Nevertheless, we proceed to identify population structures that serve as amplifiers of selection under both Bd-updating and dB-updating.

The amplifiers we identify in this work have several interesting features. First, they are robust in the sense that they amplify selection under both the Bd-updating and the dB-updating. Second, they provide amplification for any mutant fitness advantage $r$ in a range $r \in (1, 1.2)$, which covers many realistic values of the mutant fitness advantage, and the amplification is non-negligible (for instance, for $r = 1.05$ the fixation probability increases by 14% and 44%, respectively. see Fig 3). Third, the amplifiers are modular. That is, they consist of two large chunks that serve as building blocks and that interact rarely. For definiteness, in this work we

specified the two chunks and their relative sizes, but each chunk can be replaced by an alternative building block and the relative sizes can be altered. For example, the best currently known dB-amplifiers amplify by a factor of 1.5× for $r \approx 1$ and continue to amplify for $r$ in a range $r \in (1, \varphi)$, where $\varphi = \frac{1}{2}(\sqrt{5} + 1) \approx 1.618$ is the golden ratio [32]. If better dB-amplifiers are found, they can be used as a building block in place of one of the chunks to improve the range $r \in (1, 1.2)$ for which the resulting structure amplifies in both worlds.

In this work, our objective was to increase the fixation probability of an invading mutant in both worlds (Bd-updating and dB-updating). An interesting direction for future work is to optimize other quantities in both worlds.

One such quantity is the duration of the process until fixation occurs [37–39]. For example, achieving short fixation times in combination with increasing the fixation probability does not appear to be easy. Our proofs rely on the existence of small edge weights to separate the time scales at which different stages of the process happen. While using more uniform edge weights might still lead to the same outcome, the proofs would need to become more delicate. A possible approach to identify structures that serve as fast amplifiers in both worlds would be to find unweighted amplifiers, because then the time would be guaranteed to be at most polynomial [40, 41]. The first step in this direction would be to identify large and substantially strong unweighted dB-amplifiers. There are promising recent results in this direction [31].

Looking beyond fixation time, there are other relevant quantities such as the recently introduced rate at which beneficial mutations accumulate [42]. Existing research suggests that the two versions of the Moran process behave quite differently in terms of the fixation probability [21], but quite similarly in terms of the fixation time [40, 41]. Which of those two cases occurs for other relevant quantities remains to be seen.

## Supporting information

**S1 Appendix. Supplementary information for Amplifiers for both Birth-death and death-Birth updating.**
(PDF)

## Acknowledgments

We thank Gavin Rees for helpful discussions.

## Author Contributions

**Conceptualization:** Josef Tkadlec, Krishnendu Chatterjee.

**Formal analysis:** Jakub Svoboda, Soham Joshi, Josef Tkadlec.

**Funding acquisition:** Josef Tkadlec, Krishnendu Chatterjee.

**Investigation:** Jakub Svoboda, Soham Joshi.

**Software:** Josef Tkadlec.

**Supervision:** Josef Tkadlec, Krishnendu Chatterjee.

**Validation:** Josef Tkadlec.

**Writing – original draft:** Jakub Svoboda, Soham Joshi, Josef Tkadlec.

**Writing – review & editing:** Jakub Svoboda, Josef Tkadlec, Krishnendu Chatterjee.

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
