## [Decision Letter · Decision Letter 0]

27 Feb 2024

Dear Mr Svoboda,

Thank you very much for submitting your manuscript "Amplifiers of selection for the Moran process with both Birth-death and death-Birth updating" for consideration at PLOS Computational Biology. As with all papers reviewed by the journal, your manuscript was reviewed by members of the editorial board and by several independent reviewers. The reviewers appreciated the attention to an important topic. Based on the reviews, we are likely to accept this manuscript for publication, providing that you modify the manuscript according to the review recommendations.

Sincerely,

Roger Dimitri Kouyos

Academic Editor

PLOS Computational Biology

Zhaolei Zhang

Section Editor

PLOS Computational Biology

Reviewer's Responses to Questions

**Comments to the Authors:**

Reviewer #1: This is a very well written article that presents a graph structure that amplifies the selection for Bd and dB updating. The result is relatively counterintuitive, because, as the authors present in the first part of the manuscript, there are many indications that such a structure should not exist.

Overall, the proof is sound and the presentation is excellent.

Reviewer #2: This manuscript considers the question of whether a single graph can be an amplifier of selection for both Birth-death and death-Birth updating. This is a very natural question, to the extent that it is surprising no one has investigated it before.

The manuscript presents both negative and positive results. On the negative side, it proves results that help explain why such a simultaneous amplifier would be difficult to find (and has not been found previously). On the positive side, it provides a recipe for constructing such an amplifier, and proves its amplification properties.

The manuscript is straightforward and clear. I can find no issues with the proofs. Overall I think this is a positive contribution. I have only a few minor comments:

Author Summary: “Interestingly, none of the spatial structures that work for one updating seem to work for the other one.” I would rephrase to something like “so far, no spatial structures have been found that…”

Main Text:

*I do not think a research paper needs “spoiler alerts”

*Theorem 1 is presented as a “black box” in the main text, but it is really quite easy to understand: For dB, neutral fixation probability is directly proportional to (weighted) degree, while for Bd, it is inversely proportional. A vertex cannot be above average in both degree and inverse degree. Unless I am missing something, this is the entire content of Theorem 1. Of course, the rigorous proof from the AM-HM inequality is still valuable, but the intuition is very straightforward and can be presented in the main text.

SI:

*The content of Lemma 1 in some sense goes all the way back to Antal, Redner, & Sood (“Evolutionary Dynamics on Degree Heterogeneous Graphs”, PRL, 2006). The extension to weighted graphs has already been noted; see, for example, Eq. (95) of Allen & McAvoy (“A mathematical formalism for natural selection with arbitrary spatial and genetic structure”, JMB, 2018).

*It may be helpful to repeat the definition of gamma before the statement of Theorem 2. In fact, the proof of Theorem 2 starts by developing formulae for these gammas, but these could be presented before the theorem statement.

*When the proof of Theorem 2 turns to dB, it gives expressions for p_{Bd,r}^+ and p_{Bd,r}^-. I believe these should be dB rather than Bd.

*Provide a reference for Markov’s inequality

*The paragraph after the proof of Lemma 6 does not seem necessary in light of Theorem 1 (which has already been proven by this point).

*In the proof of Lemma 10, the periods after conditions 1, 2, and 3 are a bit awkward; consider removing these.

Reviewer #3: The authors explore evolutionary processes occurring on structured populations, considering two types of stochastic processes, the Birth-death and the death-Birth processes. The authors focus on the ability of networks to amplify selection of beneficial mutations under the two processes. They identify that the graph topologies previously inspected did not amplify selection for both processes at the same time. In this context, they proposed to do two things: 1) building an argument of why simultaneous amplification is not commonly observed; 2) building a network topology that is an exception to this.

To address the first point, the authors provide 3 negative results. The first empirically shows that instances of known amplifiers of selection for each of the dynamics are not amplifiers for the other. The second result shows that under neutral selection, fixation starting from one particular node of a network can never be amplified for both dynamics (Theorem 1). The third result shows that it is not possible for a node of a network to hold an amplified initial forward bias under both dynamics (Theorem 2).

To address the second point, the authors conceptualize a graph that is an amplifier of beneficial mutations under both dynamics, and prove under which conditions they do it (Theorems 3 and 4). This graph is robust, modular and does the simultaneously amplification for reasonable areas of the fitness parameter space. Due to the modularity of the constructed graph, it can be easily altered to incorporate stronger Bd- or dB- amplifiers which may be found in the future.

Overall, I consider the paper to provide a significant contribution to the study of evolutionary processes under population structure. Its main strength is the several fronts used to build the main argument of the paper which are provided with exact mathematical formalism and formalized as theorems. Particularly, negative results 2 and 3, and the positive result are entirely supported by the proofs provided in the supplementary material (Theorems 1-4 and auxiliary Lemmas). Most lemmas/theorems provided are quite general (e.g., Theorems 1 and 2). The only exception is the first negative result, which is provided without an extensive exploration of previously found amplifying topologies.

Taking this into consideration, I recommend only minor revisions as stated below.

Minor revisions needed in main manuscript:

Fig. 2 – This figure needs improvement in visualization. Colors used for K11 and D11 are too similar, and markers used for those two cases completely overlap in figures b) and c). Size of markers on b) and c) is too small and should be increased for better visualization. Alternatively, you may want to consider redoing both plots as rho_r/rho_{K11,r} in order to avoid the confusion between the curves of D11 and K11 and because, ultimately, you compare rho_r under each topology with rho_{K11,r}.

Fig. 2 - The caption states “suppressor D11”, shouldn’t it be “amplifier D11” instead?

Theorems 1 and 2 – “Then either” seems to imply only one of the three statements can be true. I would suggest rephrasing as “Then at least one of the following is true:…” to avoid ambiguity.

Page 7, line 219 – The temperature of a node u is introduced as the sum of the reciprocal of the degree of all neighbours of u.

- It should be instead the sum of w(v,u)/deg(v). This is only equal to 1/deg(v) when graphs are unweighted.

- The sum should be over “v:(v,u)\\in E“ and say it explicitly in the equation to avoid confusion, instead of explaining later only in text.

- The notation deg(v)=sum_u w(v,u) should be defined beforehand, as it is not defined anywhere in the main manuscript, only in the supplementary material.

Fig. 3 caption – “is a Fan graph on f nodes.” should be “is a Fan graph with f nodes.”

Fig. 4 – In a), I suggest representing the other edge attached to u and v for exactness of figure. In the caption, it is said “a mutant who has fixated over A^{dB}”, where it should be A^{Bd}.

Revisions needed in Supplementary Information:

Page 1- First paragraph – In “deg(u)=sum….”, the sum should be over v:(u,v)\\in E.

Page 2 – Lemmas 2 and 3 – For consistency, “N” (not “n”) should have been used for the size of the graph.

Page 2 – Theorems 1 and 2 – Similarly to what I suggested for the main text, “Then either” seems to imply only one of the three statements can be true. I would suggest rephrasing as “Then at least one of the following is true:…” to avoid ambiguity.

Page 2 – third paragraph from end – In “the right-hand side”, it should say instead “the left-hand side” or, alternatively, “the product above”.

Page 4 – First equations should be $p_{dB,r}^+=…$ and $p_{dB,r}^-=…$. They should have factors of 1/N instead of factors 1/F.

Page 4 - What is meant by "For the equality to occur everywhere"?

Page 5 - In Proof of Lemma 6, it is said “later we prove”, but it has been proved before.

Page 5 – Right before stating Lemma 7, you have “spread the the other graph”, whereas it should be “spread to the other graph”

Page 5 – Lemma 7 doesn’t read easily. I would suggest rephrasing to have the equation at the end: “… and a randomly placed mutant in $A_{N,\\gamma}$, mutants become extinct or fixate on their part of $A_{N,\\gamma}$ before any reproduction over edge $v,v’$ with probability of at least: <math xmlns="http://www.w3.org/1998/Math/MathML"><semantics><mn>1</mn><mo>-</mo><mi>O</mi><mo>(</mo><mn>1</mn><mo>/</mo><msup><mi>N</mi><mn>2</mn></msup><mo>)</mo><annotation encoding="LaTeX">1-O(1/N^2)</annotation></semantics></math>.”

Page 6 – “The sum of these probabilities is at most $2\\epsilon/N^7$”. Where was the factor of $2$ obtained from?

Page 8 – “…from Lemma 8 (if the edge is used).” should mention Lemma 9 instead.

Page 8 – “mutants do no fixate in “A^{dB}_{\\gamma N}” should instead mention A^{Bd}_{(1-\\gamma) N}”.

Page 8/9 – When stating theorems 3 and 4, there is a repetition of “graph $A_{N,\\gamma}$,” in the text that should be removed.

Page 9 – There are dot product symbols missing between $1.00001X$ in three lines of the last equation.

**Have the authors made all data and (if applicable) computational code underlying the findings in their manuscript fully available?**

Reviewer #1: None

Reviewer #2: Yes

Reviewer #3: Yes

PLOS authors have the option to publish the peer review history of their article (what does this mean?). If published, this will include your full peer review and any attached files.

Reviewer #1: No

Reviewer #2: **Yes: **Benjamin Allen

Reviewer #3: No

Figure Files:

Data Requirements:

Reproducibility:

References:

---

## [Editor Report · Decision Letter 1]

18 Mar 2024

Dear Mr Svoboda,

We are pleased to inform you that your manuscript 'Amplifiers of selection for the Moran process with both Birth-death and death-Birth updating' has been provisionally accepted for publication in PLOS Computational Biology.

Best regards,

Roger Dimitri Kouyos

Academic Editor

PLOS Computational Biology

Zhaolei Zhang

Section Editor

PLOS Computational Biology

---

## [Editor Report · Acceptance letter]

24 Mar 2024

PCOMPBIOL-D-24-00099R1 

Amplifiers of selection for the Moran process with both Birth-death and death-Birth updating

Dear Dr Svoboda,

I am pleased to inform you that your manuscript has been formally accepted for publication in PLOS Computational Biology. Your manuscript is now with our production department and you will be notified of the publication date in due course.

With kind regards,

Zsofia Freund
